# Diflunisal and Analogue Pharmacophores Mediating Suppression of Virulence Phenotypes in *Staphylococcus aureus*

**DOI:** 10.3390/antibiotics12071180

**Published:** 2023-07-12

**Authors:** Liana C. Chan, Hong K. Lee, Ling Wang, Siyang Chaili, Yan Q. Xiong, Arnold S. Bayer, Richard A. Proctor, Michael R. Yeaman

**Affiliations:** 1Division of Molecular Medicine, Harbor-UCLA Medical Center, Torrance, CA 90502, USA; lchan@lundquist.org (L.C.C.); hlee@lundquist.org (H.K.L.); lwang@lundquist.org (L.W.); 2Division of Infectious Diseases, Harbor-UCLA Medical Center, Torrance, CA 90502, USA; yxiong@lundquist.org (Y.Q.X.); abayer@lundquist.org (A.S.B.); 3Department of Medicine, David Geffen School of Medicine at UCLA, Los Angeles, CA 90024, USA; 4Institute for Infection and Immunity, The Lundquist Institute for Biomedical Innovation at Harbor-UCLA Medical Center, Torrance, CA 90502, USA; 5Vanderbilt Eye Institute, Vanderbilt University Medical Center, 2311 Pierce Ave., Nashville, TN 37232, USA; siyang.chaili@vumc.org; 6Departments of Medical Microbiology & Immunology and Medicine, University of Wisconsin School of Medicine and Public Health, Madison, WI 53705, USA; rap@wisc.edu

**Keywords:** diflunisal, analogues, virulence, pharmacophore, *Staphylococcus aureus*, antimicrobial

## Abstract

Invasive methicillin-resistant *Staphylococcus aureus* (MRSA) infections are leading causes of morbidity and mortality that are complicated by increasing resistance to conventional antibiotics. Thus, minimizing virulence and enhancing antibiotic efficacy against MRSA is a public health imperative. We originally demonstrated that diflunisal (DIF; [2-hydroxy-5-(2,4-difluorophenyl) benzoic acid]) inhibits *S. aureus* virulence factor expression. To investigate pharmacophores that are active in this function, we evaluated a library of structural analogues for their efficacy to modulate virulence phenotypes in a panel of clinically relevant *S. aureus* isolates in vitro. Overall, the positions of the phenyl, hydroxyl, and carboxylic moieties and the presence or type of halogen (F vs. Cl) influenced the efficacy of compounds in suppressing hemolysis, proteolysis, and biofilm virulence phenotypes. Analogues lacking halogens inhibited proteolysis to an extent similar to DIF but were ineffective at reducing hemolysis or biofilm production. In contrast, most analogues lacking the hydroxyl or carboxylic acid groups did not suppress proteolysis but did mitigate hemolysis and biofilm production to an extent similar to DIF. Interestingly, chirality and the substitution of fluorine with chlorine resulted in a differential reduction in virulence phenotypes. Together, this pattern of data suggests virulence-suppressing pharmacophores of DIF and structural analogues integrate halogen, hydroxyl, and carboxylic acid moiety stereochemistry. The anti-virulence effects of DIF were achieved using concentrations that are safe in humans, do not impair platelet antimicrobial functions, do not affect *S. aureus* growth, and do not alter the efficacy of conventional antibiotics. These results offer proof of concept for using novel anti-virulence strategies as adjuvants to antibiotic therapy to address the challenge of MRSA infection.

## 1. Introduction

*Staphylococcus aureus* is responsible for significant morbidity and mortality worldwide. The ability of *S. aureus* to cause diverse infections ranging from soft tissue abscesses to life-threatening invasive disease is mediated by the coordinated expression of an extensive repertoire of virulence factors including toxins, proteases, and immune avoidance effectors. Moreover, the rapid emergence of clinical isolates exhibiting multi-drug resistance, including methicillin (MRSA)- and vancomycin (VRSA)-resistant *S. aureus*, has accelerated the urgent need for novel strategies to counter *S. aureus* pathogenic mechanisms. Therefore, targeting such virulence determinants with potential novel therapeutics is highly logical and may reveal improved methods for preventing or treating infections caused by this or other human pathogens.

We originally demonstrated that diflunisal (DIF) strongly attenuates *S. aureus* virulence gene expression and phenotypes [1,2,3,4,5]. This advance extended the observations of our group and others regarding the beneficial effects of aspirin on *S. aureus* infection [6,7,8,9,10,11,12,13]. Specifically, aspirin and its primary metabolite salicylic acid appear to reduce the severity and progression of clinical and experimental infective endocarditis (IE) [6,7,13,14,15,16,17] and significantly decrease the risk of *S. aureus* bacteremia in catheterized patients [18,19]. Likewise, meta-analyses revealed a significant beneficial effect of aspirin and reduced risk of patient systemic embolism in IE [20]. Several studies have suggested that salicylates may attenuate virulence through interactions with *S. aureus* global regulatory systems [1,2,6,11,21,22,23,24]. Others have since validated these findings in a number of experimental models [25,26,27].

The current work builds upon our previous findings to explore bioactive pharmacophores of DIF and structural analogues for their ability to modulate hemolysis, proteolysis, and biofilm formation as prototypic virulence phenotypes in *S. aureus*. The present findings reveal that specific chemical moieties and their three-dimensional stereochemistries are integral to the suppression of virulence phenotypes in clinically relevant *S. aureus* isolates. These results substantiate our strategic approach to develop novel therapeutic agents to interfere with essential virulence mechanisms and thereby promote host defenses against *S. aureus*. This approach is likely to be extendable to other high-priority antibiotic-resistant pathogens and may hold promise for addressing the threat of untreatable infections.

## 2. Results

In the present study, we investigated the structure–activity relationships of DIF and structural analogues to suppress virulence phenotypes in vitro. The goal of this study was to evaluate compounds for anti-virulence efficacy relative to DIF and discern key pharmacophores that may be correlated with the inhibition of specific virulence phenotypes in *S. aureus*.

### 2.1. Selection of Staphylococcus aureus Strains

Prior studies by our group and others showed that DIF, a difluorinated analogue of salicylic acid, can mitigate the expression of virulence factors in *S. aureus* [14]. DIF and 26 structurally distinct DIF analogues were compared for their inhibition of virulence phenotypes in a panel of highly relevant *S. aureus* study strains (Table 1).

### 2.2. Selection of DIF Analogues

DIF comprises three moieties: (1) a benzoic acid aromatic ring; (2) a difluorophenyl group; and (3) a hydroxyl group (Figure 1A). The commercially available DIF analogues that were selected included representatives with differing halogen, carbonyl, or hydroxyl groups and positioning of chemical moieties (Figure 1). To determine the impact of halogens on DIF activity, hydroxyphenylbenzoic acid analogues were evaluated (Figure 1B,C; **OHPB**). Monofluorophenyl (Figure 1D; **FPB**) and difluorophenyl (Figure 1E–J; **dFPB**) isotypes were selected to assess the importance of the halogen number and positioning on the phenyl group in the absence of a hydroxyl group. Alternatively, difluorophenyl analogues lacking a carboxylic acid group were evaluated to determine the role of benzoic acid versus phenol groups (Figure 1K,L; **dFPP**). To assess the role of the halogen type in DIF efficacy, dichlorophenyl-substituted analogues (Figure 1M–S; **dCPB**) were studied. Finally, to assess the impact of the phenyl ring of DIF, halogen-benzoic acid analogues were evaluated. 

The components that may be important for structure–activity relationships are summarized in Table 2. Together, this library of compounds was used to assess the impact on the efficacy of four different anti-staphylococcal antibiotics and the mitigation of three virulence phenotypes in eight highly relevant clinical and laboratory *S. aureus* strains. Importantly, across the range of concentrations tested in the virulence phenotyping studies, neither DIF nor its analogues alone inhibited the growth of prototypical *S. aureus* strains in vitro (Appendix A).

### 2.3. Impact of DIF and Analogues on Antibiotic Susceptibility

DIF and its analogues were tested with gold-standard antibiotics for interactions that may alter *S. aureus* susceptibility or resistance. Overall, the MICs of vancomycin (VAN), daptomycin (DAP), ciprofloxacin (CIP), and rifampicin (RIF) were equivalent against *S. aureus* in the presence or absence of DIF or analogues in vitro (Figure 2). Thus, DIF and its analogues are not antagonistic to traditional antistaphylococcal therapeutics and appear to act independently from their mechanisms.

### 2.4. Impact on Virulence Phenotypes

**Hemolysis.** Consistent with DIF, its structural analogues reduced hemolysis, proteolysis, and biofilm production in *S. aureus* strains (Figure 3, Figure 4 and Figure 5). DIF inhibited hemolysis in *S. aureus* strains by 42–88% compared with the no-compound control (Figure 3). The positioning of the phenyl and hydroxyl groups in **OHPB1** resulted in similar effects to DIF, but removing the fluorides from DIF abrogated hemolysis inhibition (**OHPB2**) in most *S. aureus* strains. Furthermore, the monofluorophenyl analogue (**FPB**) was less efficacious than DIF. In contrast, the difluorophenyl benzoic acid (**dFPB**) analogues showed differential effects on hemolysis depending on the localization of fluorides on the phenyl group or carboxylic acid group on the benzene ring moiety. The analogues dFPB1, -2, and -4 showed similar effects on hemolysis compared to DIF (54–90%); however, dFPB3, -5, and -6 were less efficacious. Difluorophenyl phenols (**dFPP**; DIF analogues lacking the carboxylic acid group) showed similar inhibitory effects to DIF. Likewise, dichlorophenyl benzoic acid (**dCPB**) isotypes had differential impacts on hemolysis compared with DIF. The analogues dCPB2, -3, and -4 were better at mitigating hemolysis compared with other analogues (19–58%). The analogue dCPB2 exceeded DIF in hemolysis suppression (Figure 3; Appendix A). Results of hemolysis assays performed using sheep vs. rabbit blood were highly concordant (Appendix A).

**Proteolysis**. Proteolysis assays demonstrated that DIF completely inhibited detectable proteolysis production in all strains (Figure 4). Likewise, hydroxyphenyl benzoic acid (**OHPB**) analogues completely inhibited proteolysis in most study strains. In contrast, fluorophenyl benzoic acid and phenol analogues (**FPB**; **dFPB**; and **dFPP**) were ineffective in suppressing proteolysis production. These results indicate the role of the presence of both hydroxyl and carboxylic acid groups on the benzene ring in anti-proteolysis efficacy. Interestingly, dichlorophenyl benzoic acid analogues (**dCPB**) exhibited differential effects on proteolysis in *S. aureus* strains depending on phenyl- and carboxylic acid group localization. For example, dCPB1, dCPB2, and dCPB3 are analogues with the same 3,5-dichlorophenyl group, but this group is in different carbon positions relative to the carboxylic acid group on the benzene ring. Isotype dCPB1 (carbon position 2) was less efficacious than dCPB2 (carbon position 3) or dCPB3 (carbon position 4), suggesting a stereochemical role of the phenyl vs. carboxylic acid groups in the inhibition of *S. aureus* proteolysis. In comparison, the chlorinated analogues dCPB4 and dCPB5 are similar to dCPB3, whereas dCPB6 is similar to dCPB2 but with different chlorine positions. These analogues were less efficacious than DIF against COL, LAC, MW2, C15, and C16 strains. These findings suggest a stereochemical role of chlorine atoms in mitigating proteolysis in these strains. Similar to hemolysis, halogen-benzoic acid analogues did not reduce proteolysis in any *S. aureus* strain (Figure 4; Appendix A).

**Biofilm.** Next, DIF and its analogues were assessed for their ability to mitigate biofilm production as this virulence phenotype is essential for human *S. aureus* infections, including infective endocarditis, prosthetic joint infection, and device-related infection [33,34,35,36,37,38]. As expected, DIF inhibited biofilm production in *S. aureus* strains by 9–73% compared with the no-compound control (Figure 5). In contrast, non-halogenated hydroxyphenyl benzoic acid (**OHPB**) analogues were not as effective in reducing biofilm production compared with DIF. These results suggest an important role of halogen(s) in the suppression of this virulence phenotype. Most fluorophenyl benzoic acid and phenol analogues (**FPB**; **dFPB**; and **dFPP**) exhibited strain-specific efficacy in attenuating biofilm production in laboratory and community *S. aureus* strains. However, the VISA, DNSA, and DSSA strains were generally not susceptible to biofilm inhibition by these compounds. Moreover, the analogue dFPB3 was ineffective in reducing biofilm production in most study strains (83–145%), and some study compounds (e.g., **OHPB2**, **dCPB5**, and **dCPB6**) induced biofilm formation. Similarly, dichlorophenyl benzoic acid (**dCPB**) analogues also exhibited strain- and stereo-specific efficacy in reducing biofilm production. The analogues dCPB2, dCPB3, and dCPB4 had similar or better efficacy in mitigating biofilm production than DIF, whereas closely related analogues dCPB1, dCPB5, dCPB6, and dCPB7 were worse than DIF. Similar to other virulence outcomes, halogen-benzoic acid analogues were ineffective in preventing biofilm production (Figure 5; Appendix A).

Collectively, the current results reveal structural and chirality patterns in compounds that suppress *S. aureus* virulence phenotypes. Thematically, the compounds containing a hydroxyl-diflurophenyl motif (e.g., **DIF**; Figure 6A) or containing a dichlorophenyl motif in the presence or absence of a hydroxyl moiety (e.g., **dCPB2**; Figure 6B) exhibited the greatest efficacy in suppressing hemolysis, proteolysis, and biofilm production in *S. aureus* in this study. Integrating these results, the consensus pharmacophores for the broadest anti-virulence efficacy against *S. aureus* appear to be the [hydroxy]-phenyl-benzoate and [meta]-bis-halo-phenyl moieties, which have conformational degrees of freedom associated with chiral specificity (Figure 6C).

## 3. Discussion

Conventional in vitro metrics that are used to determine the efficacy of anti-staphylococcal compounds have historically been based on growth inhibition (e.g., minimum inhibitory concentration (MIC)) or killing (e.g., minimum bactericidal concentration (MBC)) of the target organism. In the present study, we evaluated compounds for their ability to interfere with virulence factor expression, independent of growth inhibition or killing. This strategy has the advantage of avoiding the selection of antibiotic resistance and thus represents an attractive approach to innovative antimicrobial therapy.

Originally, we demonstrated the efficacies of aspirin, salicylates, and DIF against *S. aureus* in a variety of in vitro and in vivo experimental models [1,2,3,4,5,6,7,14]. Our and other subsequent studies furthered these findings regarding DIF and related compounds [39,40,41,42,43,44,45]. The present study assessed the structural determinants of DIF and its chemical analogues regarding (1) the inhibition of virulence phenotypes and (2) the impact on conventional antibiotic efficacy against relevant MRSA isolates representing diverse genotypes correlated with resistance phenotypes. Analogues differing in structure were found to demonstrate differential efficacy in inhibiting virulence phenotypes of specific *S. aureus* strains. Consistent with our previous findings, DIF was effective at inhibiting hemolysis, proteolysis, and biofilm formation in a variety of *S. aureus* strains in vitro.

The current findings verified that DIF inhibits key virulence phenotypes in *S. aureus* strains with MRSA, VISA, or DNSA resistance phenotypes representing community-acquired and healthcare-associated isolates. Thus, DIF served as a logical standard for the comparative evaluation of structurally related compounds. Relative to DIF, its analogues exerted structure-specific inhibition of the prototypic virulence phenotypes, including hemolysis, proteolysis, and biofilm production. For example, the dichlorophenyl benzoic acid compound (dCPB2) was superior to DIF in mitigating hemolysis but equivalent in its inhibition of proteolysis and biofilm formation in the majority of *S. aureus* strains tested. Interestingly, other dichlorophenyl analogues (dCPB3 and dCPB4) displayed a pattern of outcomes similar to DIF. This pattern suggested that the stereochemistry and position of the carboxylic acid and halogen moieties contribute to virulence inhibition activity. Moreover, removing halogens (F) from DIF resulted in a hydroxyphenyl benzoate (OHPB) analogue that was equal to DIF in mitigating proteolysis but did not inhibit hemolysis or biofilm production in the *S. aureus* study strains. This finding points to the importance of fluorine in inhibiting hemolysis and biofilm production but not proteolysis. In comparison, a distinct group of difluorophenyl benzoic acid (dFPB) analogues lacking a hydroxyl moiety exhibited poor inhibition of all virulence phenotypes relative to DIF or dCPB2. This result indicates the importance of the hydroxyl residue in coordination with halogen and carboxylic acid moieties for virulence suppression. The positioning of the halogens on the [meta]-bis-halo-phenyl group in relation to the positioning of the carboxylic acid or hydroxyl moieties of the [hydroxy]-benzoic acid group influences the anti-virulence efficacy of the study compounds. This finding indicates that preferential or isomer-specific stereochemical optima exist for compounds that exhibit anti-hemolysis, anti-proteolysis, and anti-biofilm efficacies. This result further implies that *S. aureus* has cognate ligands that serve as specific target(s) of these compounds.

The current findings support our hypothesis that pharmacophores of DIF and structurally related analogues interfere with essential virulence factor regulation and effector mechanisms. Interestingly, efficacious compounds consistently contained a combination of carboxylic acid and hydroxyl moieties approximating the lactone ring in autoinducing peptides. For example, the carbonyl-oxygen structure in DIF and dCPB2 appears to be integral to the virulence-suppressing mechanisms of these compounds. This concept is further supported by recent observations that DIF and structurally related hydroxyphenyl benzoate analogues (OHPB) interfere with quorum sensing and attenuate virulence factor expression in *S. aureus* [25,26,27,46]. Thus, the current findings suggest that structural similarities or common structure–activity relationships exist between chemical moieties in the pharmacophores of the study compounds and those of *S. aureus* auto-inducing peptides.

We recognize the limitations of this study. First, in vitro experiments do not recapitulate the full spectrum of host–pathogen interactions that occur during infection in vivo. Beyond the scope of the current investigation, we are actively exploring the in vivo efficacies of DIF and selected structural analogues alone as adjuvants to conventional antibiotics in highly innovative experimental models of *S. aureus* infection. Second, the current study focused only on the inhibitory effects of DIF and analogues on hemolysis, proteolysis, and biofilm formation. We appreciate that other mechanisms of virulence are also important for infection. Finally, this study included eight different *S. aureus* genetic backgrounds to represent the majority of isolates encountered in clinical infection. However, there may be other backgrounds that respond differently to the study compounds. Studies within our laboratories are ongoing to address these limitations, further define the molecular and cellular mechanisms of virulence inhibition, and evaluate strategic anti-virulence strategies for their ability to benefit conventional antibiotic efficacy in *S. aureus* infections. Collectively, the present findings and their application may accelerate innovative anti-infective strategies to meet the challenge of antibiotic-resistant MRSA infection. Such strategies may also be generalizable to address the growing threat of other high-priority human pathogens that are increasingly resistant to traditional therapies.

## 4. Materials and Methods

### 4.1. Staphylococcus aureus Strains

This study utilized 8 different well-characterized and prototypic clinical and laboratory *S. aureus* strains representing antibiotic-susceptible and -resistant genotypes and phenotypes (Table 1). Organisms from virulence-validated master cell banks were cultured to log-phase in brain–heart infusion (BHI) medium at 37 °C. The resulting cells were harvested, washed in phosphate-buffered saline (PBS; pH 7.2), sonicated, quantified with spectrophotometry, and diluted to the desired inoculum in PBS buffer.

### 4.2. Antibiotics and Study Compounds

Vancomycin (VAN), ciprofloxacin (CIP), rifampicin (RIF) (Sigma Aldrich, St. Louis, MO, USA), and daptomycin (DAP) (Merck, Branchburg, NJ, USA) were dissolved in double-distilled water (ddH20). Diflunisal (DIF) (Sigma) and its structural analogues (Combi-Blocks, San Diego, CA, USA; Sigma Aldrich, Rockville, MD, USA) (25 μg/mL; Table 2; Figure 1) were dissolved in DMSO and diluted to appropriate concentrations in aqueous buffer.

### 4.3. Growth Rate

To assess the impact of the compounds on *S. aureus* growth, strains were cultured in the presence of a study compound across a logical dose range encompassing that approved for DIF human therapy. As a reference standard for these studies, 10^6^ colony-forming units (CFU) of prototypic strains, SH1000 and LAC, were inoculated into 10 mL of tryptic soy broth, and the optical density (OD_600_) was measured hourly over 8 h (Appendix A). At the concentrations used in virulence phenotyping studies (25 μg/mL), no compound inhibited the growth of any study strain.

### 4.4. Minimum Inhibitory Concentration

The minimum inhibitory concentrations (MICs) of the antibiotics against the *S. aureus* strains were determined using the recommended standard Clinical Standards Laboratory Institute (CLSI) broth microdilution protocol [31,32]. The *S. aureus* strains were cultured in cation-adjusted Mueller–Hinton broth in the presence or absence of an antibiotic and/or study compound. The cultures were incubated overnight at 37 °C, and the lowest concentration inhibiting growth was recorded as the MIC. Assays were repeated a minimum of 3 times (*n* = 3+) on different days for experimental validation.

### 4.5. Hemolysis Assay

The strains were grown in brain–heart infusion broth (BHI; Becton Dickenson, Holdrege, NJ, USA) and inoculated (10^6^ CFU/10 μL) via microdrop plating onto tryptic soy agar containing 5% rabbit blood with or without compounds. The plates were incubated at 37 °C for 24 h followed by cold shock at 4 °C for 48 h. The diameters (mm^2^) of the zones of clearance were measured, and the data are represented as percentages of the control (*n*o compound) (Figure 3; Appendix A). Significant differences between DIF and the study compounds were analyzed using Student’s *t*-test (the *p*-values are presented in Appendix A). The values highlighted in green indicate significantly decreased (*p* < 0.05) virulence phenotypes compared with DIF, whereas values highlighted in red indicate significantly increased (*p* < 0.05) virulence phenotypes. Rabbit blood agar assays were verified in parallel with sheep blood agar plates as described above and showed comparable results (Appendix A).

### 4.6. Proteolysis Assay

The strains were grown in BHI and plated (10^6^ CFU/10 μL) as described above onto casein agar with or without compounds. The plates were grown at 37 °C for 24 h. Zones of clearance (mm^2^) were measured and normalized to the control as detailed for the hemolysis assays above. The data are represented as percentages of the control (*n*o compound) (Figure 4; Appendix A). Significant differences between DIF and the study compounds were analyzed using Student’s *t*-test as detailed above (the *p*-values are presented in Appendix A).

### 4.7. Biofilm Assay

The strains were grown in BHI with 0.5% glucose with or without DIF or an analogue for 18 h at 37 °C in 96-well flat-bottom polystyrene plates (Fisher Scientific, Waltham, MA USA). After removing the culture suspensions, the plates were washed with 1X PBS and dried at 37 °C for 1 h. Biofilms were stained with 0.1% safranin, washed with distilled water, and decolorized with 30% glacial acetic acid. The resulting biofilm densities were then measured using spectrophotometry at OD_490_ and normalized to the control. The data are represented as percentages of the control (*n*o compound) (Figure 4; Appendix A). Significant differences between DIF and the study compounds were analyzed using Student’s *t*-test as detailed in the hemolysis section (the *p*-values are presented in Appendix A).

### 4.8. Statistical Analyses

Bivariate differences in the experimental results were compared using Student’s *t*-test. The data are represented as means ± standard deviation. *p*-values of <0.05 were considered statistically significant. Statistical analyses were implemented and graphs were generated using the GraphPad Prism software.

## Figures and Tables

**Figure 1 antibiotics-12-01180-f001:**
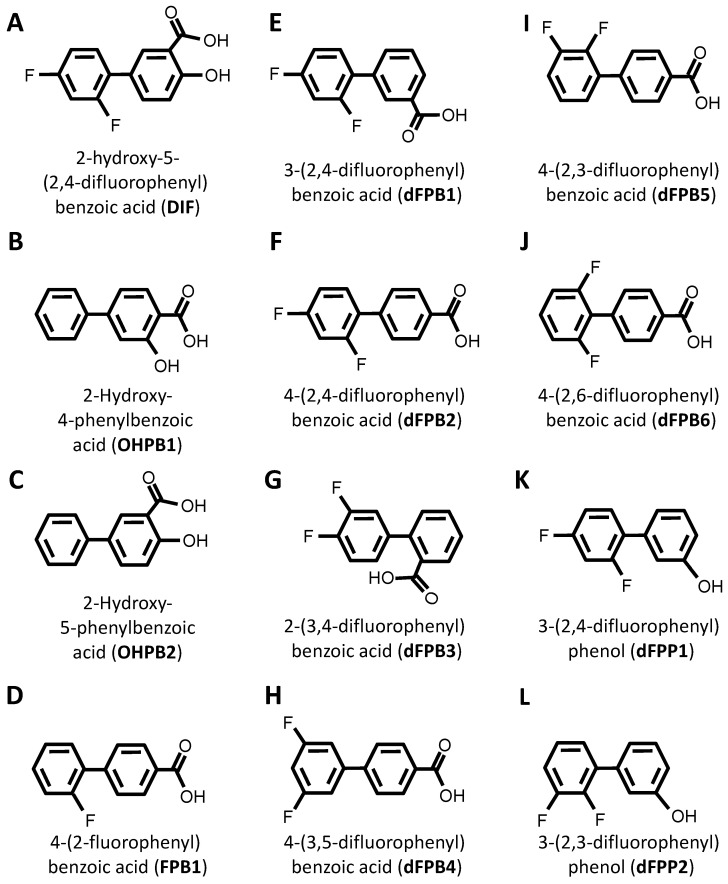
Diflunisal and comparative analogue chemical structures. Diflunisal (DIF) (**A**) and its logical structural analogues (**B**–**AA**) were compared for efficacies in mitigating virulence phenotypes. Structural analogues were selected based on (1) halogen type (fluorine vs. chlorine); (2) benzoic acid or phenol aromatic rings; (3) presence or absence of carboxyl group; (4) presence or absence of hydroxyl group; and (5) positioning of halogen, carboxyl, or hydroxyl moieties with respect to relational chirality.

**Figure 2 antibiotics-12-01180-f002:**
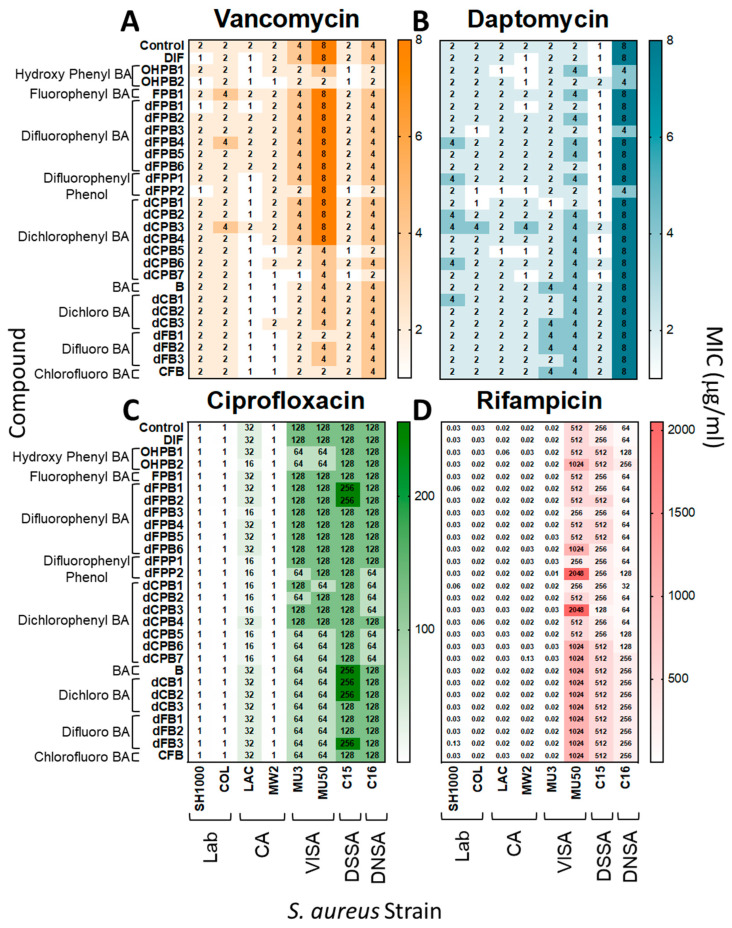
Diflunisal and structural analogues do not impact antibiotic efficacy against *S. aureus*. Minimum inhibitory concentrations (MICs) of vancomycin, daptomycin, ciprofloxacin, and rifampicin were determined for *S. aureus* strains in the presence or absence of DIF or analogues. Overall, the addition of compounds with antibiotics did not significantly alter susceptibility profiles of *S. aureus* isolates (≤2-fold difference, which is consistent with CLSI guidelines [31,32]). The data shown are representative values of three separate experiments with equivalent outcomes.

**Figure 3 antibiotics-12-01180-f003:**
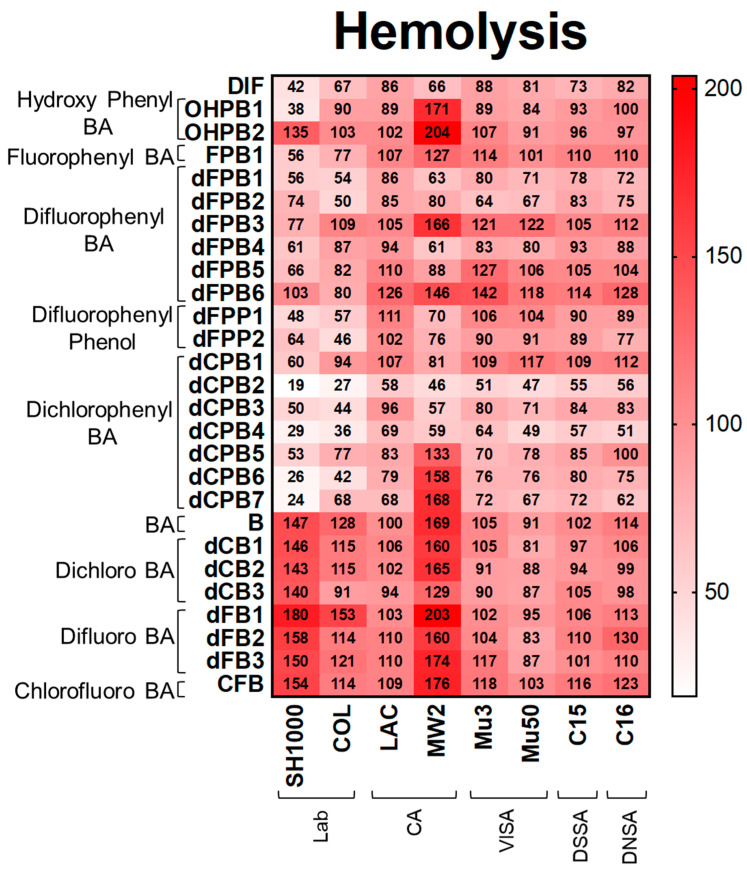
DIF and structural analogues mitigate hemolysis phenotypes of *S. aureus* study strains. Hemolysis assays were performed by plating *S. aureus* on rabbit blood agar. Values are expressed as percentages of control and represented as means of triplicate experiments (*n* = 3). Statistical analyses (*p*-value and standard deviation for each value) can be found in Appendix A.

**Figure 4 antibiotics-12-01180-f004:**
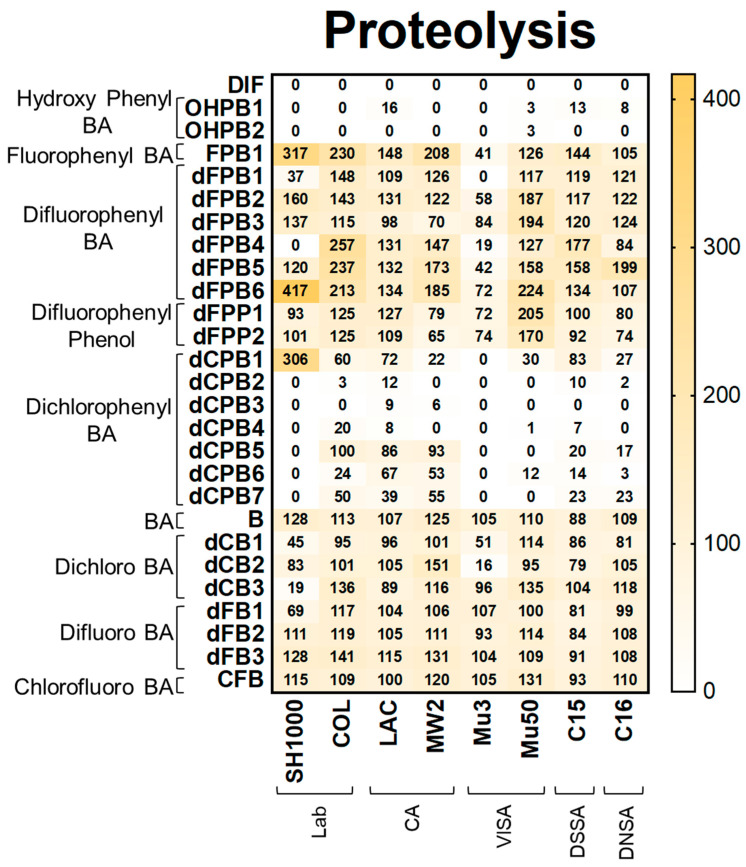
DIF and structural analogues mitigate proteolysis phenotypes in *S. aureus* study strains. Proteolysis assays were performed by plating log-phase cultures of *S. aureus* on casein agar. Values are expressed as percentages of control and represented as means of triplicate experiments (*n* = 3). Statistical analyses (*p*-value and standard deviation for each value) can be found in Appendix A.

**Figure 5 antibiotics-12-01180-f005:**
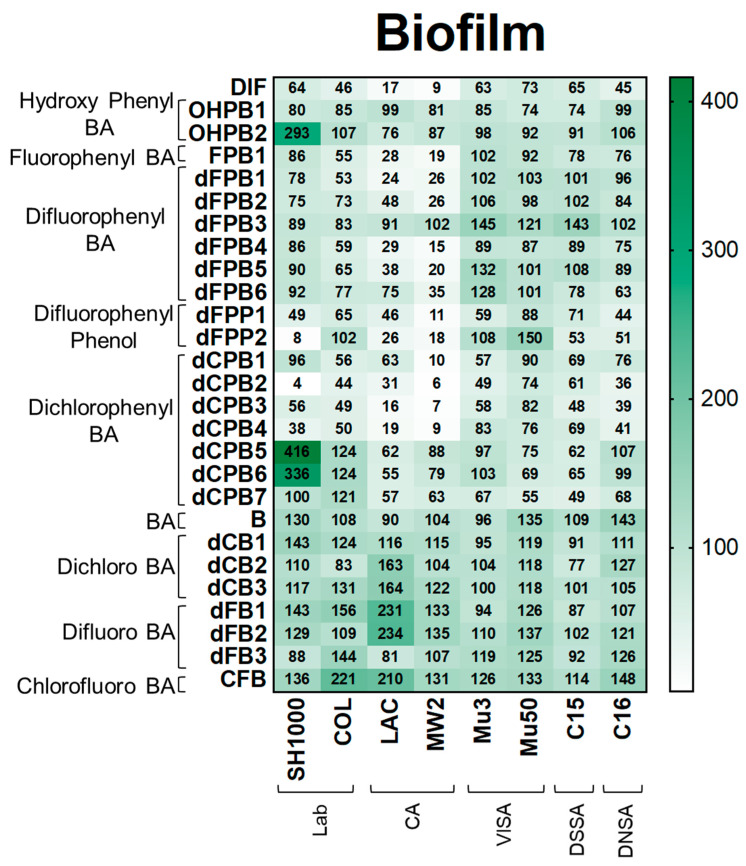
DIF and structural analogues mitigate biofilm production in *S. aureus* study strains. Biofilm assays were performed by growing log-phase cultures of *S. aureus* in brain-heart infusion broth with glucose. Cultures were removed and resultant biofilms were stained with safranin. Values are expressed as percentages of control and represented as means of triplicate experiments (*n* = 3). Statistical analyses (*p*-value and standard deviation for each value) can be found in Appendix A.

**Figure 6 antibiotics-12-01180-f006:**
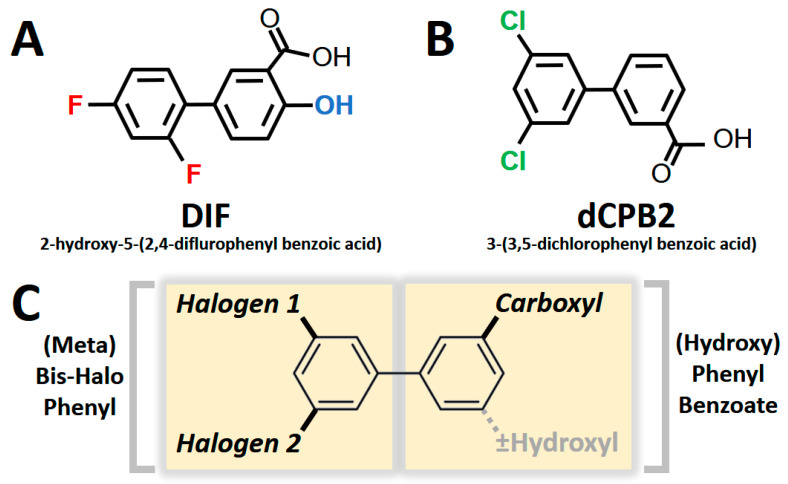
Pharmacophores correlating with broad anti-virulence efficacy against *S. aureus*. Difluorinated DIF (**A**) and a dichlorinated structural analogue dCPB2 (**B**) exhibited the greatest anti-hemolysis, anti-proteolysis, and anti-biofilm efficacy overall. These compounds are used as representatives of study compounds containing core and variable pharmacophores. Two core structures comprise a consensus pharmacophore signature (**C**): a [hydroxy]-benzoic acid group (with or without a hydroxyl moiety) and a [meta]-bis-halo-phenyl group.

**Table 1 antibiotics-12-01180-t001:** *Staphylococcus aureus* strains used in this study.

Strain	Description	Reference
SH1000	Laboratory strain 8325-4 with repaired *rsbU* mutation	American Type Culture Collection
COL	Original MRSA reference strain	American Type Culture Collection
LAC	CA-MRSA USA300 isolate; Los Angeles County	[28]
MW2	CA-MRSA USA400 isolate	[29]
Mu3	Vancomycin-intermediate MRSA (VISA)	[30]
Mu50	Vancomycin-intermediate MRSA (VISA)	[30]
C15	Daptomycin-susceptible MRSA (DSSA)	Present study
C16	Daptomycin-non-susceptible MRSA (DNSA)	Present study

**Table 2 antibiotics-12-01180-t002:** Compounds used in this study.

Name	Compound Name	M.W	Phenyl Group (Position)	Halogen (Position)	Aromatic Compound
DIF	2-hydroxy-5-(2,4-difluorophenyl) benzoic acid	250.2	Difluorophenyl (5)	FI (2, 4)	Benzoic acid
OHPB1	2-Hydroxy-4-phenylbenzoic acid	214.2	None (4)	NA	Benzoic acid
OHPB2	2-Hydroxy-5-phenylbenzoic acid	214.2	None (5)	NA	Benzoic acid
FPB1	4-(2-fluorophenyl) benzoic acid	216.2	Fluorophenyl (4)	FI (2)	Benzoic acid
dFPB1	3-(2,4-difluorophenyl) benzoic acid	234.2	Difluorophenyl (3)	FI (2, 4)	Benzoic acid
dFPB2	4-(2,4-difluorophenyl) benzoic acid	234.2	Difluorophenyl (4)	FI (2, 4)	Benzoic acid
dFPB3	2-(3,4-difluorophenyl) benzoic acid	234.2	Difluorophenyl (2)	FI (3, 4)	Benzoic acid
dFPB4	4-(3,4-difluorophenyl) benzoic acid	234.2	Difluorophenyl (4)	FI (3, 5)	Benzoic acid
dFPB5	4-(2,3-difluorophenyl) benzoic acid	234.2	Difluorophenyl (4)	FI (2, 3)	Benzoic acid
dFPB6	4-(2,5-difluorophenyl) benzoic acid	234.2	Difluorophenyl (4)	FI (2, 6)	Benzoic acid
dFPP1	3-(2,4-difluorophenyl) phenol	206.2	Difluorophenyl (3)	FI (2, 4)	Phenol
dFPP2	3-(2,3-difluorophenyl) phenol	206.2	Difluorophenyl (3)	FI (2, 3)	Phenol
dCPB1	2-(3,5-dichlorophenyl) benzoic acid	267.1	Dichlorophenyl (2)	Cl (3, 5)	Benzoic acid
dCPB2	3-(3,5-dichlorophenyl) benzoic acid	267.1	Dichlorophenyl (3)	Cl (3, 5)	Benzoic acid
dCPB3	4-(3,5-dichlorophenyl) benzoic acid	267.1	Dichlorophenyl (4)	Cl (3, 5)	Benzoic acid
dCPB4	4-(2,3-dichlorophenyl) benzoic acid	267.1	Dichlorophenyl (3)	Cl (2, 5)	Benzoic acid
dCPB5	4-(2,5-dichlorophenyl) benzoic acid	267.1	Dichlorophenyl (4)	Cl (2, 3)	Benzoic acid
dCPB6	2-(3,5-dichlorophenyl) benzoic acid	267.1	Dichlorophenyl (4)	Cl (2, 5)	Benzoic acid
dCPB7	3-(2,4-dichlorophenyl) benzoic acid	267.1	Dichlorophenyl (3)	Cl (2, 4)	Benzoic acid
B	Benzoic acid	122.1	NA	NA	Benzoic acid
dCB1	2,4-Dichlorobenzoic acid	191.0	NA	NA	Benzoic acid
dCB2	3,4-Dichlorobenzoic acid	191.0	NA	NA	Benzoic acid
dCB3	2,5-Dichlorobenzoic acid	191.0	NA	NA	Benzoic acid
dFB1	2,6-Difluorobenzoic acid	158.1	NA	NA	Benzoic acid
dFB2	2,3-Difluorobenzoic acid	158.1	NA	NA	Benzoic acid
dFB3	2,4-Difluorobenzoic acid	158.1	NA	NA	Benzoic acid
CFB	2-Chloro-4-fluorobenzoic acid	174.6	NA	NA	Benzoic acid

## Data Availability

The data are contained within this article (Figure 2, Figure 3, Figure 4 and Figure 5) and the Appendix A.

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
