# Peer review of "Diflunisal and Analogue Pharmacophores Mediating Suppression of Virulence Phenotypes in Staphylococcus aureus"

_antibiotics, 2023, doi:10.3390/antibiotics12071180_

Round 1
Reviewer 1 Report
The authors presented their findings in a clear and concise way. The summary of structures and results in the SI is very helpful to the readers. I recommend to accept the manuscript without any changes.
Author Response
We thank Reviewer 1 for their most favorable recommendation.
Author Response
Characterization and engineering studies of a new endolysin from the Propionibacterium acnes bacteriophage PAC1 for the development of a broad- spectrum artilysins with altered specificity
The reviewer would like to thank the authors for submitting the manuscript. Overall, the article is well defined. However, some major revisions are required to improve manuscript quality.
The response above is curious and suggests the reviewer comments are not entirely relevant.
If the responses below pertain to the current manuscript, specific responses are provided:
- S. aureus make the bacterial scientific name italic throughout the manuscript.
We thank the reviewer for their attention to detail. The manuscript has been updated as suggested.
- Lines 305-307; At concentrations used in virulence studies > Move these lines to the result section.
We appreciate this comment and have modified the statement in the results section to emphasize this point: Lines 108-110.
- Line 309; against study S. aureus > against S. aureus.
The manuscript has been updated. Please see line 315. Thank you.
- Lines 82–98; Authors have claimed to analysed the functional group of the compounds used in this study, but authors have not mentioned the method or softwares to analyze the structures of the compounds. If these structures are already known then there is no need to put this figure in the manuscript.
Thank you for this suggestion. We have clarified the point that analogues were evaluated to assess comparative influences of structural moieties.
We strongly prefer to retain the figure illustrating comparative difference among study compounds as it makes interpretation of results more meaningful as emphasized by Reviewers 1 & 3.
- In the manuscript the authors have tested “Diflunisal and Analogue Pharmacophores” against S. aureus. These compounds are commercially available and have not synthesized in present study. To check the effects of these compounds against one species members and publishing data is not a scientific approach. The data is not enough to be published. The author need to improve and extend the data like need to check these compounds against multiple pathogenic strains, need to develop new compounds or check the effects of these compounds used in present study against S. aureus members and provide mechanism through transcriptomics data or other omics data, by providing SEM images of bacterial cells disruption and make a sound story by comparing compounds with already available antibiotics.
We appreciate the Reviewer opinion. However, we wish to point out that many of the points raised are already addressed in the paper. For example, as the reviewer states, we have indeed already assessed herein the comparative effects of study compounds on 8 pathogenic S. aureus strains of distinct genotypic and phenotypic backgrounds that are highly relevant to the goals of the study. Moreover, our companion publication (Chan et al., Antibiotics 2023) did in fact assess transcriptional impact of diflunisal and comparative compounds. Therefore, we respectfully believe that the relevant points in this reviewer comment are substantively addressed in the present manuscript, in alignment with comments of Reviewers 1 & 3. The suggestion that we synthesize novel molecules is beyond the scope of the current study.
Reviewer 3 Report
Peer review
antibiotics-2435590
Diflunisal and Analogue Pharmacophores Mediating Suppression of Virulence Phenotypes in Staphylococcus aureus
Overall comments:
The authors have previously reported diflunisal (DIF; [2-hydroxy-5-(2,4-difluorophenyl) benzoic acid]) exhibiting anti-virulence factor properties. In this study, they obtained analogues with different functional groups compared to the parental compound to determine structural activity relationships. The manuscript evaluates the effect of different moieties on hemolysis, proteolysis and biofilm. Halogen, hydroxyl and carboxylic acid moieties were found to have different effects on different aspects of the compound’s properties. The analogues displayed low toxicity to both platelets and bacteria. Low toxicity to S.aureus will reduce the probability of resistance development, allowing this compound to used as a adjuvant to conventional antibiotic therapy.
Specific comments:
· Please review the manuscript’s spelling and grammatical.
o Please italicize “S.aureus” where needed in the manuscript.
· Please cite the CLSI manual for MIC broth microdilution in the methodology.
o “CLSI, M07--Methods for Dilution Antimicrobial Susceptibility Tests for bacteria that grow Aerobically, 11th Edition.”
· Please confirm what volume of bacterial culture was microdrop plated in the hemolysis assay.
· Was any additional toxicity testing done with mammalian cells lines such as HEPG2 or Hela or A549 etc?
· Were any of the previously tested mRNA expression of virulence factors tested with DIF conducted using a subset of these analogues?
Please check over the manuscript for spelling and grammatical errors.
Author Response
Please italicize “S. aureus” where needed in the manuscript.
We thank the reviewer for their attention to detail. The manuscript has been updated as suggested.
Please cite the CLSI manual for MIC broth microdilution in the methodology.
“CLSI, M07--Methods for Dilution Antimicrobial Susceptibility Tests for bacteria that grow Aerobically, 11th Edition.”
Thank you for pointing out this important citation. The manuscript has been revised accordingly and the citation included in the revised references (Ref # 29 &30)
Please confirm what volume of bacterial culture was microdrop plated in the hemolysis assay.
Thank you for requesting this detail. We have added the specific volume of “106 CFU / 10ml” to the revised manuscript (please see lines 324 & 335).
Was any additional toxicity testing done with mammalian cells lines such as HEPG2 or Hela or A549 etc?
Thank you for this comment. As toxicity was not a direct focus of the current study, we did not perform additional toxicity testing with mammalian cell lines in the current manuscript. In a study that we are preparing for submission, analogues were tested in vivo and found to be well-tolerated and without any detectable toxicity at concentrations capable of inhibiting S. aureus virulence factor expression.
Were any of the previously tested mRNA expression of virulence factors tested with DIF conducted using a subset of these analogues?
Thank you for this comment. As with the rationale summarized above, we are preparing a manuscript for submission which examines the transcriptional impact of analogues that are comparative to DIF. As the Reviewer no doubt appreciates, these studies are extensive and beyond the scope of the current manuscript.
Reviewer 4 Report
In the manuscript ID: antibiotics-2435590 the authors compare the anti-virulence properties of Diflunisal (DIF) and structure analogue compounds, trying to identify the main motives involved in the repression of virulence factors in Staphylococcus aureus. By comparing the reference molecule to the analogues against S. aureus hemolysis, proteolysis and biofilm formation, the authors identified the presence of a hydroxyl-diflurophenyl motif/a dichlorophenyl motif as the two fundamental structures to ensure the repression of the pathogen virulent phenotypes.
The manuscript is well conceived and structured, with a good rationale. The results are supported by a good statistically analysis and are well commented in the discussion. However, there are some concerns that need to be answered before publication:
-Please provide a better quality Table 1.
-In lines 92, 93 the authors state “Components that may be important for structure-activity relationships are summarized in Supplemental Table 2”; however, this table reports the p values obtained from the DIF-analogues comparison in inhibiting proteolysis. Do the authors refer to the subsequent supplementary tables? Please explain.
-Please indicate in the Supplementary Figure 1 caption the adopted DIF/analogues concentration. Was a toxic compound (e.g., an antibiotic) used as positive control to further highlight the lack of growth inhibition?
-Please clearly explain the meaning of “2-fold difference” (line 120): do the authors mean two 2-fold dilutions? Some MIC results in presence of the compounds, in particular for and vancomycin and rifampicin, are 4-times higher/lower than that obtained in absence of the compounds.
-Although the hemolysis and proteolysis activity of DIF have been recently published (doi: 10.3390/antibiotics12050902) it is advisable to include positive control compounds (e.g., protease inhibitors) to further corroborate the reported data.
-In line 298 the authors refer to “Table 2” which is not present in the manuscript. Please explain.
After solving these major revisions, the paper can be considered for publication.
MINOR COMMENTS
Please type “S. aureus” in italic throughout the manuscript;
Please type “e.g.,” in italic throughout the manuscript;
Line 31, please substitute “levels” with “concentrations/doses”;
Author Response
-Please provide a better quality Table 1.
Thank you for this suggestion. The resolution and formatting of Table 1 have been improved.
-In lines 92, 93 the authors state “Components that may be important for structure-activity relationships are summarized in Supplemental Table 2”; however, this table reports the p values obtained from the DIF-analogues comparison in inhibiting proteolysis. Do the authors refer to the subsequent supplementary tables? Please explain.
Thank you for pointing out our inadvertent statement in the prior version of the paper. We intended to state that these data are summarized in “Table 2”, not “Supplemental Table 2”. The correction has been made in the revised manuscript (please see line 106).
-Please indicate in the Supplementary Figure 1 caption the adopted DIF/analogues concentration. Was a toxic compound (e.g., an antibiotic) used as positive control to further highlight the lack of growth inhibition?
We routinely include controls (e.g. antibiotics or other anti-infective agents) in our growth inhibition studies. As expected, anti-staphylococcal antibiotics significantly inhibit growth in comparison to DIF or analogues. We prefer to exclude controls from the graphical results for clarity, however we have added a clarification of this point in the caption of Supplemental Figure 1.
-Please clearly explain the meaning of “2-fold difference” (line 120): do the authors mean two 2-fold dilutions? Some MIC results in presence of the compounds, in particular for and vancomycin and rifampicin, are 4-times higher/lower than that obtained in absence of the compounds.
Thank you for suggesting that we clarify this point. As used in the current study, a threshold of 2-fold change in MIC was the breakpoint considered to have significance. This threshold is consistent with CLSI guidelines. This clarification has been made in the revised manuscript.
-Although the hemolysis and proteolysis activity of DIF have been recently published (doi: 10.3390/antibiotics12050902) it is advisable to include positive control compounds (e.g.,protease inhibitors) to further corroborate the reported data.
We thank the Reviewer for their important suggestion. DIF and analogues are not likely to be conventional protease inhibitors or hemolysis inhibitors. Rather, our working hypothesis is that these compounds indirectly inhibit such virulence factor expression by way of altering quorum sensing and related pathways. Therefore, use of protease inhibitors could lead to spurious interpretations in this respect. Beyond the scope of the current manuscript, we are conducting extensive analyses to understand the mechanisms by which DIF and the structural analogues achieve their anti-virulence effects. Certainly, included in this ongoing work is a relevant panel of conventional protease and hemolysis inhibitors as controls, which will be interpreted carefully in context of direct vs. indirect (quorum sensing) effects.
-In line 298 the authors refer to “Table 2” which is not present in the manuscript. Please explain.
We thank the Reviewer for their attention to detail. We inadvertently omitted Table 2 from the prior version of the manuscript. The revised manuscript now contains Table 2 as intended.
After solving these major revisions, the paper can be considered for publication.
MINOR COMMENTS
Please type “S. aureus” in italic throughout the manuscript;
The manuscript has been updated. Thank you.
Please type “e.g.,” in italic throughout the manuscript;
The manuscript has been updated. Thank you.
Line 31, please substitute “levels” with “concentrations/doses”;
We thank the Reviewer for their attention to detail. The manuscript has been updated (Lines 31, 109).
Round 2
Reviewer 4 Report
In the revised version of the manuscript ID: antibiotics-2435590, the authors have answered almost all the raised concerns, thus improving the quality of the paper and making it clearer for the reader. The only not performed revision has been motivated in a satisfactory way. There are no further concerns for the publication of the paper.